# A genome-wide association study identifies a novel susceptibility locus for the immunogenicity of polyethylene glycol

Chia-Jung Chang[1], Chien-Hsiun Chen[1,2], Bing-Mae Chen[1], Yu-Cheng Su[1], Ying-Ting Chen[1], Michael S. Hershfield[3], Ming-Ta Michael Lee[1,4], Tian-Lu Cheng[5], Yuan-Tsong Chen[1,6], Steve R. Roffler[1,7] & Jer-Yuarn Wu[1,2]

Conjugation of polyethylene glycol (PEG) to therapeutic molecules can improve bioavailability and therapeutic efficacy. However, some healthy individuals have pre-existing anti-PEG antibodies and certain patients develop anti-PEG antibody during treatment with PEGylated medicines, suggesting that genetics might play a role in PEG immunogenicity. Here we perform genome-wide association studies for anti-PEG IgM and IgG responses in Han Chinese with 177 and 140 individuals, defined as positive for anti-PEG IgM and IgG responses, respectively, and with 492 subjects without either anti-PEG IgM or IgG as controls. We validate the association results in the replication cohort, consisting of 211 and 192 subjects with anti-PEG IgM and anti-PEG IgG, respectively, and 596 controls. We identify the immunoglobulin heavy chain (*IGH*) locus to be associated with anti-PEG IgM response at genome-wide significance ($P = 2.23 \times 10^{-22}$). Our findings may provide novel genetic markers for predicting the immunogenicity of PEG and efficacy of PEGylated therapeutics.

[1] Institute of Biomedical Sciences, Academia Sinica, Taipei 11529, Taiwan. [2] School of Chinese Medicine, China Medical University, Taichung 40447, Taiwan. [3] Department of Medicine, Duke University Medical Center, Box 3049, Durham, North Carolina 27710, USA. [4] Genomic Medicine Institute, Geisinger Health System, Danville, Pennsylvania 17822, USA. [5] Department of Biomedical Science and Environmental Biology, Center for Biomarkers and Biotech Drugs, Kaohsiung Medical University, Kaohsiung 80708, Taiwan. [6] Department of Pediatrics, Duke University Medical Center, Durham, North Carolina 27710, USA. [7] Graduate Institute of Medicine, College of Medicine, Kaohsiung Medical University, Kaohsiung 80708, Taiwan. Chia-Jung Chang, Chien-Hsiun Chen and Bing-Mae Chen contributed equally to this work. Correspondence and requests for materials should be addressed to Y.-T.C. (email: chen0010@ibms.sinica.edu.tw) or to S.R.R. (email: sroff@ibms.sinica.edu.tw) or to J.-Y.W. (email: jywu@ibms.sinica.edu.tw)

Polyethylene glycol (PEG) is a linear or branched polymer with repeating ethylene oxide subunits. The conjugation of PEGs to active recombinant proteins and peptides, termed PEGylation, allows significant increase of their solubility, extension of circulation half-lives, and reduction in immunogenicity; thus improving clinical efficacy[1, 2]. Owing to the therapeutic benefits of PEGylation, more than ten PEGylated drugs are approved by the Food and Drug Administration (FDA). PEG has been considered as a biocompatible and non-antigenic material over the years[3–5].

PEG has historically been considered to be poorly immuno-genic but recent studies have demonstrated the occurrence of anti-PEG antibodies in animal models immunized with PEGylated proteins and in patients treated with PEGylated biopharmaceuticals[6–11]. In addition, PEG-therapeutics are capable of priming immune responses and inducing the accelerated blood clearance of a subsequent infusion[6, 7]. This might compromise therapeutic efficacy and safety of administered PEGylated medicines by influencing their pharmacokinetics and increasing risk of infusion reactions, thus raising questions on the clinical impact of anti-PEG antibodies[9, 12].

It is noteworthy that some patients who developed anti-PEG antibody did so after the first or second infusion of PEGylated therapeutics, whereas other patients do not develop anti-PEG antibodies after the initial infusions and appear unlikely to do so during subsequent treatments[9, 12]. Recently, a high prevalence of pre-existing anti-PEG IgM and IgG antibodies have been detected in the plasma of healthy donors[13, 14]. The existence of anti-PEG antibodies in a portion of the population suggests that genetic variants might be associated with the immunogenicity of PEG. So far, comprehensive genetic assessment of the variable antibody response to PEG is lacking. In this study, we aim to identify novel susceptibility loci that might predispose individuals toward inducing antibody responses against PEG by utilizing a genome-wide association study (GWAS) scan in a Han Chinese population residing in Taiwan and to elucidate the immunological mechanisms underlying anti-PEGylated medicine antibody production.

We identify seven single-nucleotide polymorphisms (SNPs) that reach the genome-wide significance ($P < 8.97 \times 10^{-8}$) in joint analysis and localize in V (variable) segment of immunoglobulin heavy chain (*IGH*) gene. The most significant SNP rs12590237 has a P-value of $2.23 \times 10^{-22}$ (Cochran-Armitage trend test), with odds ratio (OR) = 2.36. We demonstrate that the risk allele of rs12590237 is significantly correlated with higher prevalence and concentrations of anti-PEG IgM in the plasma. These findings shed new light on the genetic basis for the immunogenicity of PEG and reveal potential genetic markers likely used for PEGylated drug therapeutics' efficacy prediction.

## Results

**Genome-wide association analysis.** We performed two case-control GWAS in Han Chinese to identify loci associated with anti-PEG IgM and IgG, respectively, using an Affymetrix Axiom CHB1 array. We initially analyzed 628,132 SNPs in a total of 177 cases and 492 controls for GWAS of anti-PEG IgM and 140 cases and 492 controls for GWAS of anti-PEG IgG. After stringent quality control filtering and kinship analysis, 557,394 SNPs (representing 89% of array SNPs) were analyzed in the discovery stage for IgM and IgG, respectively (Supplementary

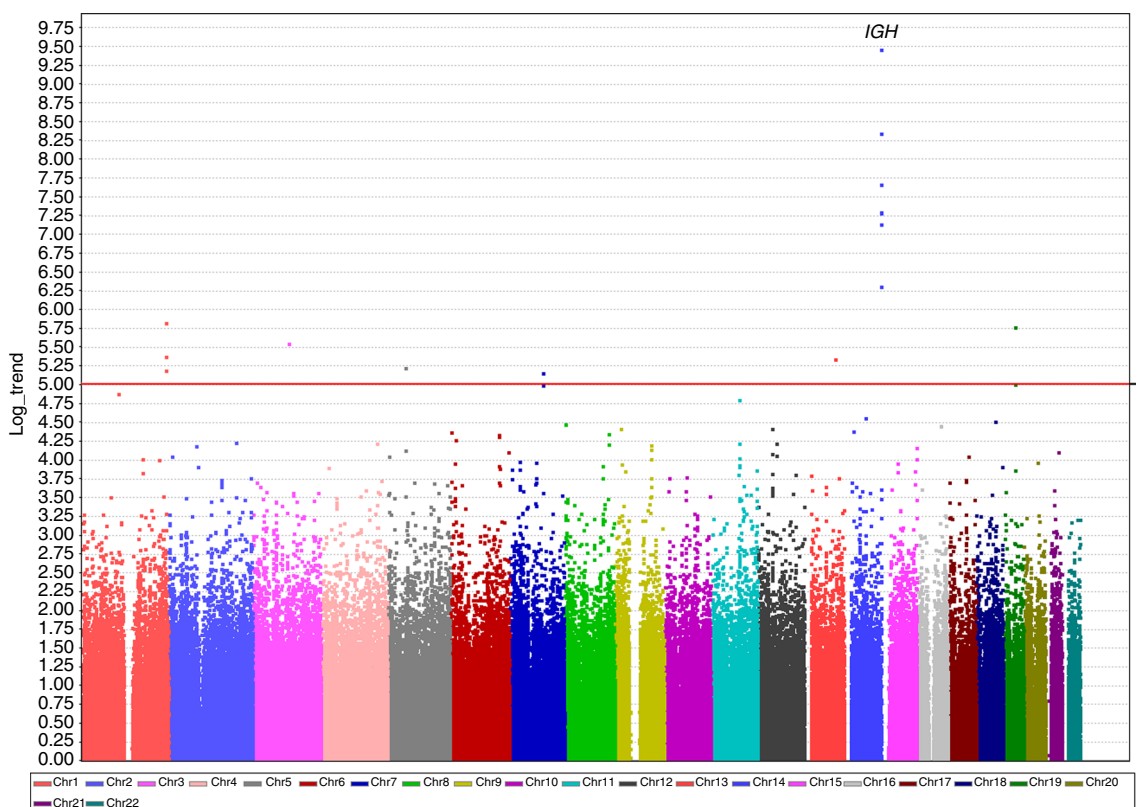

**Fig. 1** Manhattan plots for genome-wide SNPs associated with anti-PEG IgM responses. Results of genome-wide association analysis ($-\log_{10} P$) shown in chromosomal order for 557,394 SNPs tested for association in initial sample of 177 cases (anti-PEG IgM positive) and 492 controls (double-negative for anti-PEG IgM and IgG). The x axis represents each of the SNPs used in the primary scan. The y axis represents the $-\log_{10} P$-value of the trend test. Horizontal lines indicate $-\log_{10} P = 5$. Signals in the *IGH* regions are indicated

**Table 1 Association analyses of the seven SNPs achieving genome-wide significance in the joint analysis for anti-PEG IgM**

| Chr. | SNP | Position | Gene | Allele format | Risk allele | Stage | RAF controls | RAF cases | P-value | Risk allele OR (95% CI) |
|---|---|---|---|---|---|---|---|---|---|---|
| 14 | rs61999179 | 106909084 | *IGH* | AC | A | GWAS | 0.357 | 0.517 | $4.91 \times 10^{-7}$ | 1.93 (1.50–2.47) |
| | | | | | | Replication | 0.332 | 0.488 | $4.50 \times 10^{-8}$ | 1.92 (1.53–2.41) |
| | | | | | | combined | 0.338 | 0.501 | $2.13 \times 10^{-14}$ | 1.97 (1.66–2.33) |
| 14 | rs7160708 | 106944848 | *IGH* | GA | G | GWAS | 0.356 | 0.528 | $5.14 \times 10^{-8}$ | 2.03 (1.58–2.59) |
| | | | | | | Replication | 0.332 | 0.500 | $1.79 \times 10^{-9}$ | 2.01 (1.61–2.53) |
| | | | | | | combined | 0.336 | 0.513 | $5.37 \times 10^{-17}$ | 2.08 (1.76–2.46) |
| 14 | rs2157624 | 107001253 | *IGH* | AG | A | GWAS | 0.335 | 0.503 | $7.33 \times 10^{-8}$ | 2.01 (1.57–2.57) |
| | | | | | | Replication | 0.311 | 0.479 | $2.12 \times 10^{-9}$ | 2.03 (1.62–2.55) |
| | | | | | | combined | 0.316 | 0.490 | $1.71 \times 10^{-16}$ | 2.08 (1.75–2.46) |
| 14 | rs12590237 | 107001719 | *IGH* | GC | G | GWAS | 0.410 | 0.611 | $3.46 \times 10^{-10}$ | 2.27 (1.77–2.91) |
| | | | | | | Replication | 0.380 | 0.588 | $5.34 \times 10^{-13}$ | 2.33 (1.86–2.92) |
| | | | | | | combined | 0.388 | 0.599 | $2.23 \times 10^{-22}$ | 2.36 (1.99–2.79) |
| 14 | rs10143619 | 107009038 | *IGH* | CT | C | GWAS | 0.353 | 0.523 | $5.02 \times 10^{-8}$ | 2.01 (1.57–2.57) |
| | | | | | | Replication | 0.326 | 0.495 | $2.41 \times 10^{-9}$ | 2.03 (1.62–2.55) |
| | | | | | | combined | 0.332 | 0.508 | $1.13 \times 10^{-16}$ | 2.08 (1.75–2.46) |
| 14 | rs7154133 | 107022978 | *IGH* | TC | T | GWAS | 0.435 | 0.622 | $4.52 \times 10^{-9}$ | 2.14 (1.67–2.75) |
| | | | | | | Replication | 0.398 | 0.608 | $4.77 \times 10^{-13}$ | 2.35 (1.87–2.96) |
| | | | | | | combined | 0.408 | 0.615 | $2.36 \times 10^{-21}$ | 2.32 (1.95–2.74) |
| 14 | rs8007516 | 107025099 | *IGH* | CA | C | GWAS | 0.438 | 0.617 | $2.12 \times 10^{-8}$ | 2.07 (1.61–2.66) |
| | | | | | | Replication | 0.406 | 0.612 | $1.09 \times 10^{-12}$ | 2.32 (1.84–2.91) |
| | | | | | | combined | 0.414 | 0.615 | $2.60 \times 10^{-20}$ | 2.26 (1.91–2.68) |

Cases, anti-PEG IgM; Chr., chromosome; Controls, double-negative for anti-PEG IgM and IgG; Gene, Gene the SNPs located; OR, odds ratio for risk allele; $P$ = Trend $P$-value derived from a one-degree-of-freedom of Armitage trend test; RAF, risk allele frequency; 95% CI, 95% confidence interval

Table 1). Principal-component analysis (PCA) of population structure revealed no significant evidence for population structural stratification between cases and controls (Supplementary Fig. 1a, b). Quantile–quantile (Q–Q) plots were then used to examine $P$-value distributions. The genomic inflation factor was 1.049 for GWAS of IgM (Supplementary Fig. 1c) and 1.027 for IgG (Supplementary Fig. 1d).

We found genome-wide significant associations ($P < 8.97 \times 10^{-8}$; Cochran-Armitage trend test) between the immunoglobulin heavy chain (*IGH*) locus and anti-PEG IgM response in the discovery stage (Fig. 1). To validate this locus and to search for other susceptibility loci that are associated with either anti-PEG IgM response or anti-PEG IgG response, we selected 16 and 4 SNPs, respectively, with $P$-value less than $1 \times 10^{-5}$ (Cochran-Armitage trend test), for further validation and replication (Fig. 1; Supplementary Fig. 2; Supplementary Tables 2, 3). We validated these SNPs using Sequenom MassARRAY and subsequently replicated them in an independent cohort of 211 cases and 596 controls for anti-PEG IgM and 192 cases and 596 controls for anti-PEG IgG. In the replication stage for anti-PEG IgM, seven SNPs, clustered in the *IGH* locus, all showed association with anti-IgM response, and the $P$-values of those SNPs in a joint analysis ranged from $2.13 \times 10^{-14}$ to $2.23 \times 10^{-22}$ (Cochran-Armitage trend test), reaching genome-wide significance ($P < 8.97 \times 10^{-8}$; Table 1). For these SNPs, no significant heterogeneity between samples from the GWAS and the replication study was detected ($I^2 = 0$, $P_{het} > 0.5109$; Supplementary Table 4). However, none of the SNPs associated with anti-PEG IgG in the discovery GWAS had nominal $P$-values $< 0.05$ in the replication cohort (Supplementary Table 3). The genome-wide significant SNPs for anti-PEG IgM in the joint analysis mapped to chromosome 14 with physical positions ranging from 106,909,084 to 107,025,099 and located at the *IGH* gene (encoding immunoglobulin heavy chain; Fig. 2). In particular, all of the SNPs localized in the V (variable) segment of the *IGH* gene. The most significant SNP rs12590237 ($P = 2.23 \times 10^{-22}$; Cochran-Armitage trend test; odds ratio (OR) = 2.36) that directly reached genome-wide significance in the

discovery stage mapped within a 23.4-kb linkage disequilibrium (LD) block (position 107,001,719–107,025,099; Supplementary Fig. 3a). The other six SNPs with genome-wide significance are in LD with rs12590237 ($0.939 < D' < 0.996$ and $0.680 < r^2 < 0.904$; Fig. 2; Supplementary Fig. 3a). Two-point logistic regression analyses performed in any combination of rs12590237 and each linkage SNP showed that no significant associations were found in other linked SNPs (Supplementary Fig. 3b). This result indicates that these linked SNPs are not independently associated with anti-PEG IgM and associations are mainly driven by the high-LD with the most significant SNP rs12590237.

**Association of SNP rs12590237 with anti-PEG IgM responses.** Because SNP rs12590237 was most significantly associated with positive anti-PEG IgM, the influence of its different genotypes on the prevalence and levels of anti-PEG IgM were investigated. The incidence of anti-PEG IgM was significantly greater in G/G and G/C genotypes as compared to the C/C genotype (G/G, 38.3%; G/C, 32.1%; C/C, 9.6%; $P < 0.01$; z-score test for two population proportions; Fig. 3a). However, there was no significant difference in the frequency of anti-PEG IgM between G/G and G/C groups. In addition, the mean concentrations of anti-PEG IgM were highest for the homozygous risk allele of the G/G group, followed in decreasing order by the G/C and C/C groups (G/G, 2.092 µg ml$^{-1}$; G/C, 1.449 µg ml$^{-1}$; C/C, 0.724 µg ml$^{-1}$ Fig. 3b). Using nonparametric statistics with Kruskal–Wallis test, levels of anti-PEG IgM in the three genotypes showed significant differences ($P = 0.0221$). In addition, subsequent pairwise comparisons showed that individuals with genotypes G/G and G/C had significantly higher anti-PEG IgM concentrations than those with the C/C genotype (G/G vs. C/C, $P = 0.0084$; G/C vs. C/C, $P = 0.0134$; Kruskal–Wallis test). These results indicate that the occurrence of the G risk allele of rs12590237 at the *IGH* locus is associated with increased plasma levels of anti-PEG IgM.

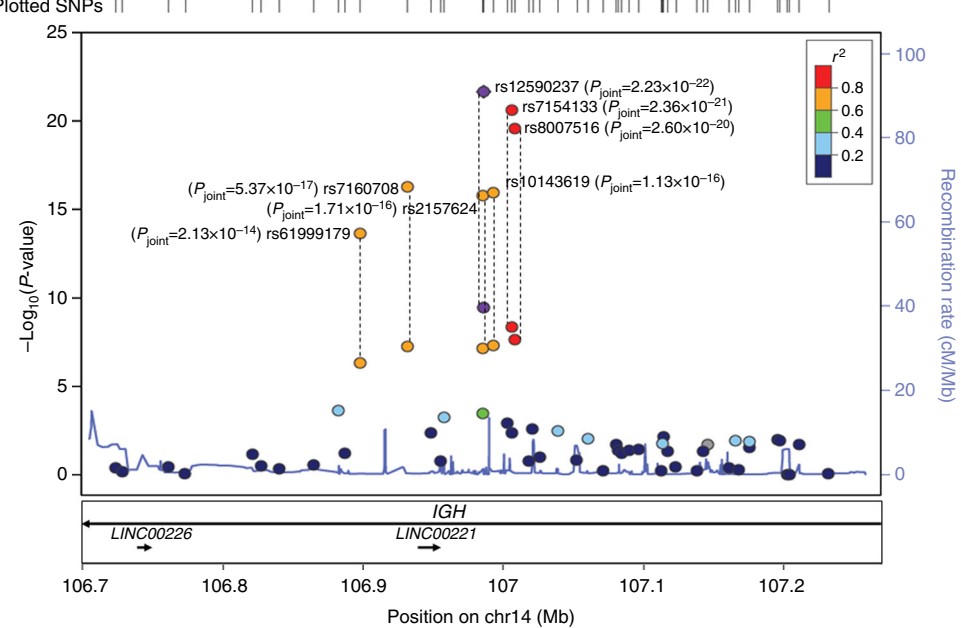

**Fig. 2** Regional association plot and linkage disequilibrium (LD) for the *IGH* loci on chromosome 14. Each SNP is plotted according to its chromosomal location (*x* axis) and its -log₁₀ *P*-values (left *y* axis) from the trend test of the primary GWAS scan and joint analysis. The dashed lines connect the same SNPs in the discovery and joint stages. The combined *P*-value ($P_{joint}$) for genome-wide significant SNP is labeled next to the dbSNP rsID. The right *y* axis and blue line showed estimated recombination rates based on the hg19/1000 genomes 2014 ASN version. SNPs are colored to reflect its LD (using the $r^2$ algorithm) with the most significant SNP (rs12590237, purple)

## Discussion

The presence of anti-PEG antibodies has been shown to affect the therapeutic efficacy and safety of PEGylated drugs[9, 11, 12], thus arousing serious concerns and stimulating interest in understanding the immunological mechanisms of anti-PEG antibody formation. In this study, we aimed to identify genetic variations that predispose individuals toward producing anti-PEG antibodies in a Han Chinese population residing in Taiwan. Most significantly, we identified 7 novel SNPs within the immunoglobulin heavy chain region (*IGH*) that reached genome-wide association significance for the presence of anti-PEG IgM. We also demonstrated that the most significant SNP rs12590237 was significantly correlated with higher prevalence and levels of anti-PEG IgM in plasma. To our knowledge, this is the first study that provides insight into the underlying genetic factors for the immunogenicity of PEG and identifies novel genetic markers that may be applicable for future therapeutic interventions. Furthermore, the G allele frequencies of the top SNP rs12590237 in 1000 Genomes reference are 0.39, 0.32, 0.41, and 0.34 for the African, American, East Asian, and European groups respectively, and similar to our control group (RAF controls = 0.388). The similar allele frequency of the top SNP across the different ethnic groups suggests that SNP rs12590237 could be considered as a universal biomarker for prediction of anti-PEG IgM responses among different ethnic groups.

The *IGH* gene is one of the three primary immunoglobulin (IG) loci of the human genome, comprised of variable (V), diversity (D), and joining (J) segments[15]. The rearranged V-D-J genes in developing B cells produce an individual's highly variable antibody repertoire that binds to a vast variety of antigens[15]. *IGHV* gene germline polymorphisms may cause amino acid substitutions that have critical functional consequences such as higher affinity and protective efficacy for antigen recognition[16]. Polymorphisms in the complementarity determining region

(CDR) of the *IGHV* locus can influence the expression of a particular IGHV repertoire[17]. In our study, all the identified SNPs associated with anti-PEG IgM response localized in the variable region of *IGH*. In the Genotype-Tissue Expression (GTEx) database of expression quantitative trait locus (eQTL) analysis, the risk allele of a significantly associated SNP (rs7154133) correlated with lower expression of *IGHV3-53* in the spleen and whole blood[18]. Therefore, we suggest that the genome-wide significant SNPs for anti-PEG IgM are associated with specific usage of an *IGHV* germline gene instead of *IGHV3-53* that is prone to express specific antibody against PEG.

Immunological memory is a biological phenomenon that establishes long-term, prophylactic humoral memory to foreign antigens[19]. The classical memory B cells in humans are switched memory B cells that express IgG, IgA, or IgE and derive from T cell-dependent (TD) responses in the germinal center (GC)[19]. However, another population of memory B cells, called IgM memory comprises about 25% of B cells in the peripheral blood[20]. IgM memory B cells, also called natural memory or natural effector memory B cells, develop in the absence of germinal centers in T cell-independent (TI) immune responses, or derive from a particular developmental pathway with somatic hypermutation (SHM)[20, 21]. The general function of IgM memory populations, in contrast to class-switched subsets, is to induce rapid and robust immune responses to foreign antigens[19]. Moreover, IgM memory subsets have been described with different origins and developmental pathways as compared to switched memory B cells because of highly significant differences in their Ig heavy chain repertoires[22, 23]. In our study, we clearly identified three anti-PEG positive populations possessing IgM only, IgG only, or double positive for IgM and IgG and one purely negative population without either anti-PEG IgM or IgG (Supplementary Table 5). This result suggests that healthy individuals exposed to PEG-modified materials, such as cosmetics, toothpaste, shampoos, or soft drinks in daily life may

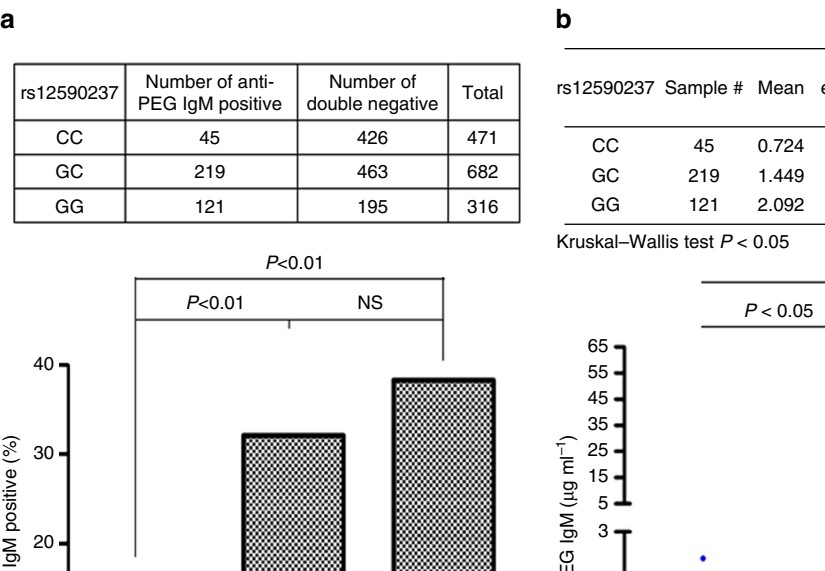

**Fig. 3** Correlation between *IGH* polymorphism (rs12590237) and antibody response of anti-PEG IgM. **a** The percentage of anti-PEG IgM positive in different genotype of SNP rs12590237. Significant differences among different genotypes were calculated using the *z*-score test for two population proportions. **b** Comparison of relative concentration of anti-PEG IgM in different genotype of SNP rs12590237. The plasma levels of anti-PEG antibodies from 1985 individuals were conducted by anti-PEG ELISA assay. The relative concentration of samples with anti-PEG IgM was calculated by comparison with cAGP4-IgM standard curves. Significant differences among different genotypes were performed by using the Kruskal–Wallis test

produce different subsets of anti-PEG memory cells. More importantly, not all anti-PEG IgM memory cells become switched to anti-PEG IgG memory cells because a portion of healthy individuals uniquely produce anti-PEG IgM antibodies under casual exposure to PEG compounds. In addition, we proved that the *IGH* locus was associated with production of anti-PEG IgM and the genome-wide significant SNPs in the *IGH* region were not in association with other healthy populations producing anti-PEG IgG. These results imply that anti-PEG IgM memory B cells and switched anti-PEG IgG B cell subsets may derive from cells with different IGH repertoires because no consistent polymorphisms in the *IGH* region were identified. Since genetic polymorphisms in the *IGH* locus may determine the origin and subset of anti-PEG antibodies, it is worth to further identify *IGH* polymorphisms associated with switched anti-PEG IgG subsets.

In clinical practice, a subset of patients with pre-existing anti-PEG antibodies can develop serious allergic reactions upon first exposure to PEGylated medicines[24]. Thus, further investigation is warranted to develop effective methods for discernment of this population before PEGylated drug administration. In this study, we have identified and replicated the variable segment of *IGH* as a novel risk locus associated with IgM antibody responses to PEG. These SNPs located at the *IGHV* gene might predispose individuals to the formation of anti-PEG antibodies and could be utilized as genetic markers to predict drug efficacy of PEGylated therapeutics and risk of allergic reactions to PEGylated drugs.

## Methods
**Study subjects**. The 885 healthy subjects in the GWAS and the 1100 healthy subjects in the replication study were randomly selected from the Taiwan Han

Chinese Cell and Genome Bank in Taiwan. Individuals range in age from 21 to 93 years (with a mean of 51 years) and they consist of 997 women and 988 men. The studies were approved by the Institutional Review Boards and the Ethics Committees of Academia Sinica in Taiwan. Written informed consent was obtained from the subjects in accordance with institutional requirements and Declaration of Helsinki principles.

**Plasma sample collection and human anti-PEG assay**. Plasma samples of healthy subjects were enrolled from a prior project that had been collected, centrifuged, and stored at the National Center for Genome Medicine, Academia Sinica[25]. All subjects of this study agreed to offer the remaining centrifugal plasma for other research in a de-linked fashion. Frozen plasma aliquots were stored at -80°C until anti-PEG antibody determinations. The human anti-PEG assay was used to determine the plasma levels of anti-PEG IgG and IgM according to our previous study[14]. Maxisorp 96-well microplates (Nalge-Nunc International, Rochester, NY) were coated with $NH_2$-$PEG_{10000}$-$NH_2$ overnight at 4 °C. Human plasma samples at dilutions of 25, 50, and 100-fold and serially diluted c3.3-IgG or cAGP4-IgM antibody standards (in duplicate) were added to separate plates at room temperature for 1 h. The horseradish peroxidase (HRP)-conjugated goat $F(ab')_2$ antihuman IgG Fc or HRP-conjugated goat $F(ab')_2$ antihuman IgM $Fc_{5\mu}$ were added to the IgG or IgM detection plates, respectively, for 1 h at room temperature. The plates were washed before adding ABTS substrate for 30 min at room temperature. The absorbance (405 nm) of wells was measured in a microplate reader (Molecular Devices). Positive responses were defined as samples with absorbance values at least 3 times greater than the mean background absorbance and more than 35% reduction of absorbance reading by PEG-liposomes competition assay. The relative concentrations of anti-PEG IgG or IgM in positive samples was calculated by comparison with c3.3-IgG or cAGP4-IgM standard curves, respectively.

**Phenotype definition of healthy subjects**. Antibodies against polyethylene glycol (PEG) were quantified in plasma samples from random selected Han Chinese residing in Taiwan by direct ELISA with confirmation by a competition assay[14]. A summary of the age and sex distribution in the GWAS and replication cohort is presented in Supplementary Table 6. The total number of anti-PEG positive and negative in the GWAS and replication cohort is presented in Supplementary Table 5. In order to get clear association signals with anti-PEG

IgM only and IgG only, we excluded subjects double positive for anti-PEG IgM and IgG from anti-PEG IgM and IgG populations, and used subjects without either anti-PEG IgM or IgG as controls. In the GWAS cohort, 177 and 140 individuals out of 885 subjects are defined as positive for anti-PEG IgM only and IgG only, respectively, and 492 individuals without either anti-PEG IgM or IgG are defined as controls. In the replication cohort with a total of 1100 subjects, there were 211 subjects positive for anti-PEG IgM only, 192 subjects for anti-PEG IgG only, and 596 subjects as controls.

**Genotyping and quality control**. Genomic DNA was extracted from blood using the Puregene DNA Isolation Kit (Gentra Systems, Inc., Minneapolis, MN). Each individual was genotyped using the Affymetrix Genome-Wide Human SNP Axiom CHB1 array (with a total of 628,132 SNPs) according to the manufacturer's protocols by the National Center for Genome Medicine (NCGM) at Academia Sinica. The call rates of all samples were greater than 95%. First-degree relatives (parent-offspring and full sibling pairs) in healthy subjects classified in case and control groups by the human anti-PEG assay were identified by kinship analysis and were excluded from further analysis. Genotyping quality control for each SNP was further evaluated by determining the total call rate (successful call rate) and minor allele frequency (MAF) in cases and controls. SNPs were excluded from further analysis if only one allele appeared in cases and controls, the total call rate was less than 0.95 or the total MAF was less than 0.05. SNPs departing significantly from Hardy-Weinberg equilibrium ($P < 1 \times 10^{-4}$) were also excluded from further analysis.

**Statistical analysis**. To access population stratification that could influence association tests, principle component analysis was carried out using EIGEN-STRAT 2.0[26]. We also estimated the variance inflation factor for genomic control. The Cochran-Armitage trend test for genome-wide association analysis was used to compare allele and genotype frequencies between cases and controls. A quantile–quantile plot was used to examine the P-value distribution. SNPs with P-values $<8.97 \times 10^{-8}$ were considered to be significantly associated with the traits. Logistic regression model of two-point analyses were used to estimate the affected status of two SNPs and their interaction. SNPs were coded as 0, 1 and 2 for the number of minor alleles and were treated as continuous variables. Heterogeneity tests ($I^2$ and P-values of the Q statistics) were performed for each SNP before combination of GWAS and replication analysis. The association tests and heterogeneity tests were performed using PLINK 1.9 (http://pngu.mgh.harvard.edu/~purcell/plink). Logistic regression was carried out using SAS software (SAS Institute, Inc., Cary, NC, USA.). The power calculation of effect size was performed using CaTS with parameters estimated from the current sample[27]: prevalence of the trait = 20%, risk allele frequency in case = 60%, and relative risk = 2.3. The sample sizes for the two-stage GWAS reached a statistical power of 99% in the joint analysis.

**Validation and replication**. The top SNPs ($P < 1 \times 10^{-5}$) from the genome-wide association analysis of anti-PEG IgM and IgG were further validated in 156 subjects using MALDI-TOF mass spectrometry (MassARRAY; Sequenom, San Diego, CA, USA).The SNP genotypes with over 96% successful rate and over 99% concordance rate between the two platforms were then genotyped in an additional independent Han Chinese replication cohort.

**Correlation analysis of genotypes and anti-PEG IgM response**. The association between genotypes of the most significant SNP (rs12590237) and anti-PEG IgM was conducted with subjects in the GWAS and replication stage. Significance of differences in positive IgM frequencies among different genotypes was calculated using the z-score calculator for two population proportions. Overall comparisons between the genotype classes and concentrations of anti-PEG IgM were performed by using the Kruskal–Wallis test. Statistical analyses were performed by using Prism 4 software (Graphpad Software, Inc.). The statistical significance was set at $P < 0.05$.

**Data availability**. The data that support the findings of this study are available from the corresponding author on request.

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

## Acknowledgements

We gratefully acknowledge the members of the Translational Resource Center (NSC102-2325-B-001-023) of the National Research Program for Biopharmaceuticals and the National Center for Genome Medicine (NSC102-2319-B-001-001) of the National Core Facility Program for Biotechnology, National Science Council, at Academia Sinica for their support in project management, subject recruitment, data coordination, genotyping performance, and statistical analysis. This study was supported by the Academia Sinica Genomic Medicine Multicenter Study (40-05-GMM), National Science Council Research grant (NSC 102-2314-B-182-053-MY3), a grant

from the Academia Sinica Research Program on Nanoscience and Nanotechnology and Translational Resource Center for Genomic Medicine (TRC) (MOST103-2325-B-001-017) of the National Research Program for Biopharmaceuticals (NRPB), Taiwan. The funders had no role in study design, data collection, or analysis, and did not influence the decision to publish or the preparation of the article.

## Author contributions

Y.-T.C., S.R.R., and J.-Y.W. obtained the financial support and conceived the study. C.-J.C., C.-H.C., and J.-Y.W. designed the experiments. C.-J.C., B.-M.C., and Y.-C.S. performed the experiments. C.-J.C., C.-H.C., Y.-T.C., B.-M.C., S.R.R., and J.-Y.W. analyzed the data. C.-J.C., C.-H.C., M.S.H., M.-T.M.L., Y.-T.C., S.R.R., and J.-Y.W. contributed to the writing of the article. All authors agree with the results and conclusions of this article. T.-L.C., Y.-T.C., S.R.R., and J.-Y.W. contributed to reagents/materials/analysis tools.

## Additional information

**Competing interests:** The authors declare no competing financial interests.

