## [Peer Review File · Nature Communications]

Description of Supplementary Files

File Name: Supplementary Information

Description: Supplementary Figures and Supplementary Tables

File Name: Peer Review File

1

2 **Supplementary Figure 1 | Principal components analysis and quantile-quantile**
 3 **plot (Q-Q plot) for the trend test.** (a,b) The EIGENSTRAT 2.0 was used to conduct
 4 principal components analysis (PCA) of the GWAS samples with HapMap
 5 populations (a) and the GWAS samples only (b) for population stratification. (c,d)
 6 Q-Q plot of the P values in Cochran-Armitage trend test. The lambda value is 1.049
 7 for anti-PEG IgM and 1.027 for anti-PEG IgG.

8

9 **Supplementary Figure 2 | Manhattan plots for genome-wide SNPs associated**

10 **with anti-PEG IgG responses.** Results of genome-wide association analysis ($-\log_{10}$

11 P) shown in chromosomal order for 557,394 SNPs tested for association in initial

12 samples of 140 cases (anti-PEG IgG positive) and 492 controls (double negative for

13 IgM and IgG). The x axis represents each of the SNPs used in the primary scan. The y

14 axis represents the $-\log_{10} P$ value of the trend test. Horizontal lines indicate $-\log_{10} P$

15 = 5.

a

IGH SNP1	IGH SNP2	D'	r^2
	rs7154133	0.994	0.904
	rs8007516	0.990	0.892
rs12590237	rs2157624	0.996	0.710
	rs7160708	0.940	0.699
	rs61999179	0.943	0.687
	rs10143619	0.939	0.680

b

SNP	P value for rs12590237	P value for linked_snp
rs61999179	1.00E-04	6.97E-01
rs7160708	6.87E-04	8.48E-01
rs12590237 rs2157624	1.07E-03	8.07E-01
rs10143619	7.05E-04	7.28E-01
rs7154133	3.74E-02	6.13E-01
rs8007516	1.01E-02	3.05E-01

16

17 **Supplementary Figure 3 | The LD structure and logistic regression analyses in**

18 ***IGH* loci.** (a) LD patterns, D' value and r^2 among the genome-wide associated SNPs

19 in *IGH* region. (b) The P -values for the rs12590237 and the linked SNPs in logistic

20 regression models conditioned on rs12590237. No significant effects of the linked

21 SNPs were found in logistic regression models conditioned on rs12590237.

22 **Supplementary Table 1. Quality control of participant data.**

23

Number at start of QC	628,132 SNPs
--------------

SNPs excluded during QC steps:	
Call rate in cases or controls <95%	12,546 SNPs
Overall MAF < 5%	42,876 SNPs
HWE for control ($P < 10^{-4}$) with polymorphic in control	4,832 SNPs

Number at end of QC	557,394 SNPs
--------------

24 HWE, Hardy–Weinberg equilibrium; MAF, minor allele frequency;

25 QC, quality control

26

27

28

29

30

31

32

33

34

35

36

37

38

39

40

41

42

43

44

45

46

47

48

49

50

51 **Supplementary Table 2. Validated SNPs (n=16) with $P_{trend} < 10^{-5}$ in the discovery**
 52 **stage of GWAS for anti-PEG IgM.**

Chr.	SNP	Position	Gene	Allele format	Risk Allele	RAF controls	RAF cases	P	Risk allele OR (95%CI)
1	rs78378504	242715847	no gene	AG	A	0.043	0.113	4.18E-06	2.819 (1.789-4.442)
1	rs75966753	242720432	no gene	CA	C	0.047	0.122	1.47E-06	2.807 (1.817-4.338)
1	rs76254089	242765102	no gene	TC	T	0.045	0.113	6.46E-06	2.716 (1.737-4.246)
3	rs75133071	105951670	no gene	AG	G	0.783	0.897	2.84E-06	2.416 (1.657-3.521)
5	rs76793186	52411735	no gene	GT	G	0.056	0.130	5.92E-06	2.506 (1.660-3.785)
7	rs740209	97653811	no gene	GT	T	0.571	0.706	6.93E-06	1.805 (1.390-2.344)
13	rs76900511	90874497	no gene	GT	T	0.631	0.767	4.50E-06	1.923 (1.454-2.541)
14	rs61999179	106909084	IGH	AC	A	0.357	0.517	4.91E-07	1.925 (1.503-2.466)
14	rs7160708	106944848	IGH	GA	G	0.356	0.528	5.14E-08	2.025 (1.582-2.592)
14	rs2157624	107001253	IGH	AG	A	0.335	0.503	7.33E-08	2.008 (1.567-2.572)
14	rs12590237	107001719	IGH	GC	G	0.410	0.611	3.46E-10	2.269 (1.767-2.912)
14	rs10143619	107009038	IGH	CT	C	0.353	0.523	5.02E-08	2.007 (1.568-2.569)
14	rs7154133	107022978	IGH	TC	T	0.435	0.622	4.52E-09	2.143 (1.669-2.751)
14	rs8007516	107025099	IGH	CA	C	0.438	0.617	2.12E-08	2.068 (1.610-2.656)
19	rs7252042	33720811	no gene	TC	T	0.154	0.257	9.81E-06	1.900 (1.415-2.551)
19	rs17694094	33723705	no gene	GT	G	0.164	0.277	1.68E-06	1.957 (1.468-2.609)

53 Chr. = chromosome; Gene = Gene the SNPs located; RAF = risk allele frequency.

54 Controls = double negative for anti-PEG IgM and IgG; Cases = anti-PEG IgM.

55 95% CI = 95% confidence interval; OR = odds ratio for risk-allele.

56 P = Trend P derived from a one-degree-of-freedom of Armitage trend test.

57

58

59

60

61

62

63

64

65

66

67

68

69

70

71 **Supplementary Table 3. GWAS and replication results in the SNPs ($P_{trend} < 10^{-5}$**
 72 **in GWAS) for anti-PEG IgG.**

Chr.	SNP	Position	Gene	Allele format	Risk allele	Stage	RAF controls	RAF cases	Trend P
2	rs2373305	38029531	no gene	TC	T	GWAS	0.321	0.496	9.22E-08
						Replication	0.369	0.404	2.32E-01
2	rs11899949	38049069	no gene	AG	A	GWAS	0.310	0.482	9.89E-08
						Replication	0.359	0.380	4.44E-01
6	rs77137655	35510259	no gene	AG	A	GWAS	0.059	0.139	4.26E-06
						Replication	0.084	0.073	5.00E-01
21	rs79714376	29197039	no gene	CT	C	GWAS	0.048	0.125	9.39E-06
						Replication	0.084	0.070	3.98E-01

73 Gene = Genes containing the SNP; RAF controls = risk allele frequency in control.

74 RAF cases = risk allele frequency in case.

75

76

77

78

79

80

81

82

83

84

85

86

87

88

89

90

91

92

93

94

95

96

97

98

99 **Supplementary Table 4. Heterogeneity test between GWAS and replication groups**
 100 **for analyses of anti-PEG IgM.**

Chr.	SNP	A1	A2	Heterogeneity Q test	I^2	Trend- P GWAS	Trend- P Replicate	Trend- P Combined
14	rs61999179	A	C	0.9758	0	4.91E-07	4.50E-08	2.13E-14
14	rs7160708	G	A	0.9723	0	5.14E-08	1.79E-09	5.37E-17
14	rs2157624	A	G	0.9448	0	7.33E-08	2.12E-09	1.71E-16
14	rs12590237	G	C	0.8795	0	3.46E-10	5.34E-13	2.23E-22
14	rs10143619	C	T	0.9516	0	5.02E-08	2.41E-09	1.13E-16
14	rs7154133	T	C	0.5886	0	4.52E-09	4.77E-13	2.36E-21
14	rs8007516	C	A	0.5109	0	2.12E-08	1.09E-12	2.60E-20

101
 102
 103
 104
 105
 106
 107
 108
 109
 110
 111
 112
 113
 114
 115
 116
 117
 118
 119
 120
 121
 122
 123
 124
 125
 126
 127

128 **Supplementary Table 5. Prevalence of anti-PEG antibodies in GWAS and**
 129 **replication samples.**

	Number of IgM positive only	Number of IgG positive only	Number of double positive	Number of double negative	Total
GWAS	177 (20%)	140 (15.8%)	76 (8.6%)	492 (55.6%)	885
Replication	211 (19.2%)	192 (17.5%)	101 (9.2%)	596 (54.2%)	1100

130
 131
 132
 133
 134
 135
 136
 137
 138
 139
 140
 141
 142
 143
 144
 145
 146
 147
 148
 149
 150
 151
 152
 153
 154
 155
 156
 157
 158
 159
 160

161 **Supplementary Table 6. Sample population characteristics with age and sex.**

Age group (years) --- GWAS stage								
	Total	20-29	30-39	40-49	50-59	60-69	70-79	≥ 80
	885 (100%)	120 (14%)	121 (14%)	157 (18%)	142 (16%)	173 (20%)	116 (13%)	56 (6%)
Male	449 (51%)	68 (57%)	57 (47%)	84 (54%)	65 (46%)	77 (45%)	69 (59%)	29 (52%)
Female	436 (49%)	52 (43%)	64 (53%)	73 (46%)	77 (54%)	96 (55%)	47 (41%)	27 (48%)
Age group (years) --- Replication stage								
	Total	20-29	30-39	40-49	50-59	60-69	70-79	≥ 80
	1100 (100%)	152 (14%)	157 (14%)	219 (20%)	212 (19%)	182 (17%)	149 (14%)	29 (3%)
Male	539 (49%)	82 (54%)	65 (41%)	109 (50%)	109 (51%)	88 (48%)	75 (50%)	11 (38%)
Female	561 (51%)	70 (46%)	92 (59%)	110 (50%)	103 (49%)	94 (52%)	74 (50%)	18 (62%)

162

163

164